# Portuguese as Heritage Language in Germany—A Linguistic Perspective

**Esther Rinke** [1],*,[†] **and Cristina Flores** [2],[†] 

1   Institute for Romance Languages and Literatures, Goethe University Frankfurt am Main,
    60629 Frankfurt, Germany
2   Centro de Estudos Humanísticos, Universidade do Minho, 4710-057 Braga, Portugal; cflores@ilch.uminho.pt
*   Correspondence: Esther.Rinke@em.uni-frankfurt.de
†   These authors contributed equally to this work.

**Abstract:** This article provides a comprehensive overview of the contribution of linguistic research on Portuguese as a heritage language in Germany to the general understanding of heritage language development. From 1955 to 1973, nearly 166,000 Portuguese migrants found work in Germany as so-called 'guest workers' (*Gastarbeiter*). Because the aim of many Portuguese migrant families was to return to Portugal, their children met relatively good conditions for the acquisition of their heritage language. Nonetheless, second-generation heritage speakers (HSs) show some linguistic particularities in comparison to monolingual Portuguese speakers in Portugal. Based on the results of previous research, we show that the following factors shape the linguistic knowledge of this group of bilinguals: (1) Restricted exposure to the heritage language may cause a delay in the development of certain linguistic structures, (2) deviations from the standard norm may be related to the lack of formal education and the primacy of the colloquial register and (3) heritage bilinguals may accelerate ongoing diachronic development. We argue that apparent effects of influence from the environmental language can often have alternative explanations.

**Keywords:** Portuguese; heritage speakers; input; heritage language development; register variation; diachronic change



## 1. Introduction

The aim of the present paper is to connect the dots of various research results on Portuguese as a heritage language (HL) in Germany and to derive from them more general insights into HL development. The authors of this paper have worked together on several projects on Portuguese as the HL over the last 8 years. Most work has targeted bilingual speakers of Portuguese and German, who are descendants from Portuguese migrant families living in Germany—so-called heritage speakers (HSs) of Portuguese. Working as team or in collaboration with other colleagues, we have scrutinized this group of speakers from various angles, by looking at children and adults, using experimental methods and spontaneous speech corpora and investigating different grammatical domains and linguistic properties. At this point of our research, we have produced a sufficient amount of data that allows us to generalize our findings in order to contribute to a better understanding of HL development and bilingualism in general.

The population of Portuguese HSs in Germany is particularly interesting for the study of HL development for several reasons. On the one hand, the conditions for the acquisition of the HL are, in general, very favourable in comparison to other HS populations. Very often, both parents are native speakers of Portuguese, and the language is therefore regularly spoken as a home language. Also, Portuguese migrants—particularly those families that came to Germany in the 1970s—usually had a return-oriented lifestyle ([Berretta Soares 2010](#)) and therefore encouraged their children to use their family language and to participate in HL classes. These second- and third-generation speakers are typical

HSs in that they acquire German as their dominant environmental language, have less input in their HL in comparison to monolingual native speakers living in Portugal and rely predominantly on colloquial input sources with less access to formal registers. Our research has shown that—as a consequence of these particular conditions on the acquisition of the HL—Portuguese second-generation migrants living in Germany generally reach native-like competence in their HL but may also show differential acquisition and knowledge of selective linguistic properties. The selection of properties and the way in which they deviate from the monolingual grammar of Portuguese reveal that: (1) Restricted exposure to the heritage language may cause a delay in the development of certain linguistic structures, (2) deviations from the standard norm may be related to the lack of formal education and the primacy of the colloquial register and (3) heritage bilinguals may accelerate ongoing diachronic development. In contrast, the cross-linguistic influence (CLI) of German may not be a decisive factor shaping the linguistic knowledge of this population of HSs. Based on our findings, we discuss each of the claims mentioned in (1)–(3) separately. Then, we connect them to each other in the Conclusion section, in which we also address the question of CLI. We start with general background information about the Portuguese community in Germany.

## 2. Portuguese Migration in Germany

### 2.1. A Historical and Demographical Overview

Currently, about 11,228,300 people living in Germany have a foreign nationality (13.5% of the residents)[1], but the proportion of population with a migration background is twice this number, reaching over 21 million people. This number includes not only people born abroad but also later generations already born in Germany and with German citizenship. A considerable part of these in-born residents with migration background corresponds to heritage speakers of their families' language of origin, the population in the focus of this article.

In these statistics, the proportion of Portuguese migrants is rather insignificant. Only 0.2% of all migrants living in Germany have the Portuguese nationality, amounting, in 2019, to 138,410 people. This number was about 115,000 in the first decade of the current millennium and has increased in the last 10 years as a consequence of the economic crisis affecting (not only) Portugal and due to the growing working mobility within Europe. The number of Portuguese citizens requesting the German nationality has also increased in the last decade but it is still a low number compared to other nationalities. In 2000, 229 Portuguese citizens acquired the German nationality, and this number increased to 760 in 2019.[2] Globally, Germany is in the seventh position in the ranking of countries with the highest number of Portuguese residents (France tops this ranking). Within the EU, Portugal is the country with the highest percentage of population living abroad: 21% of Portuguese citizens do not live in Portugal (worldwide, Portugal is 13th in the ranking of countries with the highest proportion of emigrants).

Thus, Portuguese emigration is not a recent phenomenon: It has marked Portuguese history for centuries. Portuguese emigration started in the 16th century and increased considerably from the end of the 19th century onwards (Arroteia 2001). Between 1900 and 1988, about 3.5 million people left Portugal, 25% of them illegally. The peak of emigration was reached between 1966 and 1973 (Baganha 1994).

As for Germany, a significant movement of Portuguese migration to this country started in the late 1950s, when Portuguese society was still suffering under the authoritarian regime of the 'New State' (*Estado Novo*), which ruled in Portugal between 1933 and 1974. The migration flow from Portugal to other European countries, particularly to France, increased steadily in the second part of the 20th century, mainly due to high

---

[1] Data are from the Statistisches Bundesamt (Destatis), referring to 31 December 2019. https://www.destatis.de/DE/Themen/Gesellschaft-Umwelt/Bevoelkerung/Migration-Integration/_inhalt.html.

[2] Observatório da emigração: http://observatorioemigracao.pt/np4/paises.html?id=56.

levels of poverty, which particularly affected the underprivileged social classes from rural areas. Migration was further motivated by the Portuguese Colonial War (*Guerra Colonial Portuguesa*) between 1961 and 1974.

Even while punitive sanctions were being imposed by the international community on the Portuguese regime, Portugal and Germany signed a bilateral agreement on labour recruitment in March 1964. By this time, Germany was undergoing the so-called 'economic miracle' with increasing demand for manpower in industry. This agreement, much like the one the German government signed with several other south European countries, aimed at regulating the presence of foreign labourers in German factories. Importantly, recruitment was aimed at limited-time working stays (Steinert 2014), as reflected in the term officially used to name these labourers: *Gastarbeiter* ('guest workers'). It is an important feature that characterizes the initial period of Portuguese migration to Germany that emigration was seen as temporary and the return to Portugal an achievable aim. For this reason, in the first decades, the migrant population coming from Portugal was almost exclusively male. Women and children remained in the home country (Baganha 1994). From 1955 to 1973, the number of Portuguese labourers working in German factories and at Hamburg's harbour reached almost 166,000 (Freund 2007). From the 1970s onwards, the migration flow between Portugal and Germany has been characterized by: (i) The return to Portugal of many male migrants from the first period of migration, (ii) the migration of the family members of male migrants who decided to remain in Germany and requested a license to have their families join them and (iii) new migration waves. Interestingly, in many documented cases, families migrated without their children, or only with their youngest children, leaving the older ones in Portugal with relatives (Pimentel and Martins 2016). This indicates that the return to Portugal was a lifetime goal for many Portuguese families (Pinheiro 2010). We highlight this because it helps to explain the growing demand for Portuguese classes in areas with large Portuguese communities, e.g., Hamburg.

*2.2. Portuguese Heritage Language Education in Germany*

A possible, future return came along with the need to prepare the children, who migrated with their parents or were already born in Germany, to be able to join the Portuguese school system after the return to Portugal (De Azevedo 2003). This growing need let to the establishment of the so-called *escola portuguesa* ('Portuguese school') in the 1970s in areas with growing Portuguese communities. In the first phase, these classes, officially called *Curso de Língua e Cultura Portuguesas* ('Portuguese Language and Culture Courses'), functioned as complementary afternoon schools. With no uniform legislative basis, these classes were organized by the Portuguese embassy in Bonn and their various local consulates or by the Portuguese Catholic Mission in Germany.[3]

The curriculum of the Portuguese Language and Culture Courses aimed at teaching Portuguese as native language, similarly to the subject of 'Portuguese' taught in the homeland schools, including the use of pedagogical materials identical to those used in Portugal. This indicates the extended use of Portuguese within the families and the high/native-like proficiency that characterized the second generations' heritage language competence, at least in the first periods of Portuguese migration in Germany.

From the 1980s onwards, a reorganization of the Portuguese classes took place not only in Germany, but in all countries, with a significant presence of Portuguese communities. The classes were integrated in the official network *Ensino do Português no Estrangeiro* ('Portuguese Teaching Abroad'), coordinated by the *Instituto Camões* (now called *Camões—Institute of Cooperation and Language, I.P.*), currently under the scope of the Portuguese Ministry for Foreign Affairs. Article 19 (1.e) of the Basic Law of the Portuguese Educational System from 1986 provides Portuguese-descendant children the constitutional right to attend this network. A decree from 2006 defined a new legal basis for all Portuguese classes taught abroad. These integrated not only the courses for Portuguese descendants but also

---

[3]　A case in point are the Portuguese classes founded in 1973 by the Catholic Priest Eurico Azevedo in Hamburg (De Azevedo 2003).

the classes of Portuguese as Foreign Language offered, for instance, at language centres based at universities. In 2012, extensive legal and organizational changes were made to the Portuguese Teaching Abroad network, including substantial changes to the curricula of the Portuguese Language and Culture Courses. The curriculum lost the aim of teaching Portuguese as native language because the courses now hosted children from various family backgrounds and with different levels of proficiency, including not only highly proficient first- and second-generation children, but also receptive (third- and fourth-generation) bilinguals who did not actively speak Portuguese. The official guidelines started to include the term 'Portuguese Heritage Language,' and a unified Framework of Reference for Portuguese Teaching Abroad (*Quadro de Referência para o Ensino Português no Estrangeiro/QuaREPE*, (Grosso et al. 2011)) was introduced. Similar to the Common European Framework of Reference for Languages/CEFR, it grades the students' proficiency on a scale from A1 to C2. This framework and various further guiding documents now regulate the type of teaching activities and appropriate pedagogical materials to be used in these courses.

The teachers working in the Portuguese Teaching Abroad network are recruited in Portugal. Requirements for employment are (i) A teaching degree, (ii) certified B2 knowledge of the language of the host country and (iii) passing an exam organized by the *Camões, I.P*. In the school year 2019–2020, the network served 72,244 students and employed 978 teachers all over the world, representing an investment of more than EUR 22.5 million (Mundo Português 2019). In Germany, Portuguese Heritage Language courses are offered by the official network in more than 110 schools (in the afternoon and on Saturdays), serving about 2500 students, mainly from Portuguese families, but also from other Portuguese-speaking backgrounds (e.g., Brazil, Angola, Mozambique, Cape Verde).[4]

### 3. Effects of Reduced Input on Heritage Language Development

An increasing number of studies on bilingual language development have consistently shown that the amount of language input is a key variable influencing the process of language acquisition ((Bohman et al. 2010; Gathercole and Thomas 2009; Paradis and Jia 2017; Rodina et al. 2020; Unsworth 2013), among many others). In the case of bilingual children with a migration background, whose home language differs from the majority language, i.e., HSs, variation in the quantity of language exposure at home affects the development of their heritage language. Conversely, consistent exposure to the majority language is guaranteed through schooling and socialization outside home. Moreover, not all linguistic domains and properties of the HL are equally affected by varying language input, with effects depending on their linguistic complexity and their timing of acquisition in L1 development (Tsimpli 2014).

Evidence for the correlation between extra-linguistic variables related to amount of language exposure and differential development of different properties of a heritage language comes from several studies on European Portuguese (EP) as HL, for instance, research on lexical development (Correia and Flores 2017), on the acquisition of the subjunctive (Flores et al. 2017a), on anaphora resolution (Rinke and Flores 2018) and on pronunciation (Flores et al. 2017b).

In a study involving 23 second-generation Portuguese-descendant children living in Germany, aged 6 to 11 years, Correia and Flores (2017) compared their patterns of lexical development with those of 21 monolingual EP children, both in terms of vocabulary size and of lexical diversity. In addition, the authors investigated the effects of extra-linguistic variables related to the quantity and quality of HL exposure on the bilingual children's lexical development in EP. Data was collected through a semi-spontaneous oral production task and a detailed parental questionnaire.

---

4    See the annual report from the Coordination of the Portuguese Teaching Abroad network in Germany, based at the embassy in Berlin: https://www.berlim.embaixadaportugal.mne.pt/. Note that Portuguese Heritage Language courses are also offered by several German federal states (see the list here: https://cepealemanha.wordpress.com/portugues-no-estado-alemao/).

The study revealed a number of interesting observations. As expected, the bilingual children produced fewer nouns and verbs (but the same number of adjectives) than their monolingual peers. Table 1 shows the median and interquartile range (IQR), as well as the minimum and maximum number of lexical items (i.e., absolute frequency), produced by each group in each grammatical category. A comparison of the words used by both groups showed, in addition to a considerable overlap of the lexical items, that the heritage children used a substantial amount of EP words that monolingual children living in Portugal did not use, and vice versa, revealing that bilinguals who speak EP as HL possess a rich, diverse lexical repertoire on their own, which is not a reduced subpart of the monolinguals' repertoire.

**Table 1.** Lexical distribution and variation per group of European Portuguese (EP) speakers regarding the subcorpora.

| | Nouns | | Verbs | | Adjectives | |
|---|---|---|---|---|---|---|
| | **HS** | **MS** | **HS** | **MS** | **HS** | **MS** |
| Median | 25 | 41 | 14 | 18 | 12 | 14 |
| Minimum | 10 | 27 | 0 | 9 | 2 | 2 |
| Maximum * | 46/63 | 77 | 27 | 33 | 18/25 | 23 |
| IQR | 20–36 | 32.5–55 | 9–20 | 15–25.5 | 6–13 | 8–18.5 |

Note: HS = Heritage Speakers; MS = Monolingual Speakers; IQR = Interquartile Range; * Due to the existence of an outlier in the HS group regarding the grammatical categories of nouns and adjectives, the first (the outlier's) and second highest values are presented.

Furthermore, correlation analyses between the indicator of lexical richness and various input variables, extracted from the parental questionnaires, revealed that the HSs' lexical knowledge was significantly correlated with the quantity of input and output at home as well as with the number of EP-speaking parents. This indicates that the HSs whose parents were both native speakers of EP produced a higher number of different lexical items (particularly verbs and adjectives) than those who only had one native EP-speaking parent. The finding that there is indeed an important correlation between lexical development and amount of input received at home clearly supports the idea that the HL needs to play a major role in the daily interactions within the home environment if the intergenerational transmission of Portuguese in the diaspora is to be successful. In contrast to the majority language (German, in our case), which is used in multifaceted communication contexts, the minority language, Portuguese, is considerably more vulnerable to the effects of restricted input.

Similar conclusions were reached by Flores and colleagues in a study on mood selection involving 50 EP-speaking heritage children (and adolescents), aged 6 to 16 years (Flores et al. 2017a). On the basis of an elicited production task centred on mood choice in complement clauses, this study aimed at analysing the effect of age and amount of input in the acquisition of this grammatical property.

In Portuguese, the selection of the indicative versus the subjunctive in complement clauses is basically (but not exclusively) constrained by the semantics of the matrix predicate (see e.g., (Marques 2013)). For instance, strong epistemic verbs, which express knowledge (e.g., the verb *saber* 'to know'), select the indicative, as shown in 1a, whereas weak epistemic verbs, which express a low degree of belief (e.g., *duvidar* 'to doubt'), select the subjunctive in the complement clause (see 1b).

| 1a. | O João | sabe | que | a | Maria | **vem** | hoje. |
|---|---|---|---|---|---|---|---|
| | the John | knows | that | the | Mary | comes.IND | today |

'John knows that Mary will come today.'

| 1b. | O João | duvida | que | a | Maria | **venha** | hoje. |
|---|---|---|---|---|---|---|---|
| | the John | doubts | that | the | Mary | comes.SUBJ | today |

'John doubts that Mary will come today.'

We know from previous studies (de Jesus 2014) that there is an order of acquisition of the different mood contexts and that some subjunctive contexts (e.g., with weak epistemic verbs) are acquired very late in L1 EP. Thus, the study by Flores and colleagues asked whether late structures, such as those illustrated in 1b, are particularly vulnerable in HL development and, if so, which extra-linguistic variables were the most influential.

The results showed an effect of the subjects' age at testing. In general, acquisition of the subjunctive was delayed compared to monolinguals in the age spans of 6 to 12 years, but there was a high convergence with the monolingual grammar after 12 years of age. This reveals that protracted development may be overcome in later stages of development if there is no interruption of exposure to the heritage language. Furthermore, results revealed an important role of two variables: The number of first-generation parents and the presence of older siblings. Children who had two first-generation parents and older siblings communicating in Portuguese at home showed faster acquisition of mood choice in complement clauses than children from second-generation households (see Table 2 for the results on the production of the subjunctive per age span and parents' profile).

**Table 2.** Accuracy rate in the subjunctive conditions per parent profile and age group (in %) (from (Flores et al. 2017a)).

|  | Implicative | | Non-Implicative | | Weak Epistemic | |
|---|---|---|---|---|---|---|
|  | 1st Generation Parents | 2nd Generation Parents | 1st Generation Parents | 2nd Generation Parents | 1st Generation Parents | 2nd Generation Parents |
| 6–7 years | 0 | 4.2 | 12.5 | 20.8 | 0 | 8.3 |
| 8–9 years | 35 | 6.2 | 47.5 | 12.5 | 15 | 3.1 |
| 10–12 years | 62.5 | 28.6 | 62.5 | 25 | 37.5 | 21.4 |
| 13–16 years | 78.13 | 70.83 | 96.88 | 91.67 | 56.25 | 50 |

In some cases, the reduced input that HSs receive may cause a delay in the acquisition of certain grammatical properties. One example from our work concerns the comprehension of null and overt subject pronouns in heritage Portuguese (Rinke and Flores 2018). A total of 72 informants participated in this study: 16 HSs of EP (children and teenagers) with German (Ger) as the dominant environmental language and 20 HSs of EP living in Andorra with Spanish/Catalan (Span/Cat) as the environmental languages were compared to 18 age-matched monolingually raised Portuguese children/teenagers and 18 monolingual adults. The focus was on the interpretation of null and overt subjects in intrasentential anaphoric, intrasentential cataphoric and intersentential contexts. In all contexts, null subjects were preferentially interpreted in terms of topic continuity and overt subjects as topic switch, although to differing degrees depending on the context. In example (2), the null subject in the subordinate clause (pro) was preferentially interpreted as the subject referent of the main clause (*a mãe* 'the mother,' topic continuity) and the overt subject (*ela* 'she') as being coreferential with the object of the main clause (*a avó* 'the grandmother,' topic shift) (Calabrese 1986; Carminati 2002).

2.    A mãe        cumprimentou        a avó        quando        *pro/ela*        entrou        na cozinha
      the mother        greeted        the grand-mother        when        she        entered        in-the kitchen
      'The mother greeted the grandmother when she entered the kitchen.' (Lobo and Silva 2016) (327)

Overall, the results show that all groups of monolingual and bilingual speakers differentiated between the null subject condition and the overt subject condition. In the overt pronoun condition, monolingual and bilingual children behaved alike and differed from the monolingual adults, which shows that the interpretation of overt pronouns is a complex task in monolingual and bilingual acquisition and mastered relatively late in native EP. In the null subject condition, monolingual children/teenagers performed

more adult-like than the bilinguals and seemed to be one step ahead in comparison to the bilingual children/teenagers.

This slight delay observed in the group of bilingual children does not necessarily lead to an intergenerational change concerning the use of null and overt subjects, as shown by Flores and Rinke (2020). In a corpus study on subject expression in two generations of Portuguese migrants living in Hamburg, we showed that second-generation HSs did not differ from first generation migrants with respect to the factors constraining subject realization/omission in EP in spoken language. The speakers of both generations showed very similar overall rates of subject omission (around 67%) and they revealed sensitivity to the same determining factors of subject pronoun realization/omission, namely person and number, verb type, switch reference (TC/TS) and distance of the antecedent.

The fourth study we would like to highlight in this overview is on HSs' accent (Flores et al. 2017b). The growing literature on the phonetic and phonological competence of HSs in the last decade has shown that, on the one hand, the early and continuous exposure to the family language is a strong predictor of a native-like accent. On the other hand, many studies have found differences between heritage and homeland speakers regarding their phonetic and/or phonological performance (see (Rao 2019) for an overview on Spanish). As shown by Kupisch et al. (2020), it is particularly in the heritage language (and less the majority language) that heritage speakers may develop a differential phonological system, because this language is more subject to input alternation and variation in language use.

The case we report here shows that intensive use of the home language within Portuguese migrant families effectively predicts the development of an accent that is perceived as being native-like. The study was based on a global accent rating task in which speech samples of three types of EP speakers were rated by native EP listeners, namely (i) HSs living in Germany, (ii) German L2 learners of Portuguese and (iii) monolingual speakers who never lived abroad. Following the procedure of Moyer (2004) replicated in many other studies, the ratings combined a binary judgment of the sample sounding native versus non-native with the indication of degree of certainty (with three options, certain, semi-certain, uncertain), resulting in a six-point Likert scale (where 1 indicates 'certain of native speech' and 6 'certain of non-native speech').

The results showed that the accent of most HSs was perceived as being native-like, with ratings similar to those given to the monolinguals' speech samples. In some cases, the raters were less confident in rating HSs compared to monolinguals, indicating that the accent of some HSs showed some particularities (see Figure 1).[5]

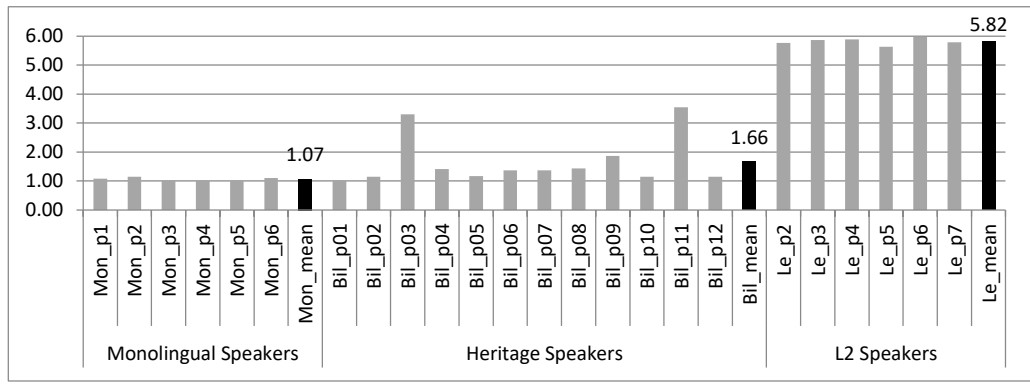

**Figure 1.** *Degree of Nativeness per Group* (from (Flores et al. 2017b)).

---

[5] A Kruskal–Wallis test corroborated that the three groups differed significantly with respect to the evaluation of their accent ($H(2) = 17.727$, $p < 0.001$). Follow-up Mann–Whitney tests with Bonferroni correction confirmed that monolinguals and L2ers differed significantly ($U = 0.000$, $p < 0.017$) and heritage bilinguals and L2 speakers did as well ($U = 0.000$, $p = 0.001$) in terms of their accent ratings. Another Mann–Whitney showed that also the comparison between heritage bilinguals and monolingual speakers revealed significant differences between the groups ($U = 5.500$, $p = 0.004$).

Unlike the studies that detect significant deviations in HSs' speech compared to native homeland speakers, the results of this study confirm a largely native-like development of Portuguese HSs' pronunciation. We explained this performance by looking at the speakers' degree of contact with Portuguese. All participants came from first-generation families, who spoke almost exclusively Portuguese at home. This turned out to be a strong indicator of native-like development in many studies, not only in the phonetic-phonological domain, but also in morphosyntax and in the lexical domain, as discussed above, despite the dominant presence of German in the speakers' lives.

## 4. The Primacy of Exposure to the Colloquial Register and the Role of Formal Instruction

Heritage speakers acquire and speak their HL predominantly in the family context, which is typically characterized by the use of the spoken and colloquial register. Differences between monolingual speakers and heritage bilinguals have, therefore, also been attributed to their lack of access to more formal registers of the HL or to the degree of literacy (in their HL), in general.

Some examples of such an approach are the studies by Rothman (2007) and Pires and Rothman (2009) on Brazilian Portuguese (BP) in the U.S., who observed that HSs of BP, in contrast to educated monolingual native speakers, lacked knowledge concerning the inflected infinitive. In accordance with studies on diachronic change that have documented a loss of inflected infinitival constructions in colloquial registers of BP (Pires 2002), the authors argued that educated monolinguals acquire the inflected infinitive through formal education and that the difference between monolingual speakers and HSs of BP can be accounted for by the lack of access to formal education in the HS group. Thus, according to the authors, BP HSs do not acquire the property because it is not in their input.

Our study on the morphosyntactic knowledge of clitics by European Portuguese HSs living in Germany also revealed an influence of colloquial input sources. The participants showed a very mixed behaviour with respect to the syntactic properties of clitic pronouns in their HL (Rinke and Flores 2014). In a grammaticality judgment task (GJT) involving different aspects of the syntax of clitics, HSs performed target-like with respect to syntactic properties which relate to the colloquial register but revealed difficulties concerning aspects which are typically acquired through formal education. We briefly exemplify this finding with respect to clitic climbing and clitic allomorphs. Clitic climbing relates to the variable position of the clitic in certain verbal complexes. As can be seen in example (3), the clitic pronoun can either occur attached to the non-finite verb (3a.) or enclitically to the finite modal verb (3b.).

| | | | | | | | |
|---|---|---|---|---|---|---|---|
| 3a. | O João | pode | comprá- | **lo** | na | semana | que | vem |
| | the John | can | buy- | CL.Acc | in-the | week | that | comes |
| | | | 'John can buy it next week.' | | | | | |
| 3b. | O João | pode | **-o** | comprar | na | semana | que | vem |
| | the John | can | CL.Acc | buy | in-the | week | that | comes |
| | | | 'John can buy it next week.' | | | | | |

This variation is truly optional in the contexts that we included in the experiment. However, in more formal registers of the language, the climbed option (3b.) is stigmatized. In general, schoolteachers of Portuguese in Portugal often correct the climbed order in favour of the non-climbed position of the clitic pronoun. In contrast, corpus studies of spoken Portuguese have revealed that clitic climbing represents the most frequent option in colloquial registers (Barbosa et al. 2017; Flores et al. 2017c). In the aforementioned GJT, the HSs had no difficulties in recognizing that both the climbed and the non-climbed order of the clitic are acceptable in constructions like (3a/b), whereas monolingual speakers often rejected the climbed order although it is grammatical. This is in accordance with its stigmatization in formal registers.

Another picture arose with respect to allomorphic clitic forms in EP. Allomorphic variants of the clitic occur in certain morphophonological contexts. One example is given

in (4), with a clitic in its (non-shaped) regular form in (4a.) and a clitic shaped by the phonological context in (4b.).

| 4a. | A Maria | viu- | **o.** | (regular form) |
| | the Mary | can | CL.Acc | |
| | | | 'Mary saw him.' | |
| 4b. | As meninas | viram | **-no.** | (allomorphic form) |
| | the girls | saw | CL.Acc | |
| | | | 'The girls saw him.' | |

Clitic allomorphy is not optional, and using the regular form of the clitic in contexts like (4b.) would be ungrammatical. However, the allomorphic form of the clitic in (4b.) is phonologically not very salient in spoken language, given that the nasal consonant [n] is attached to the nasal diphtong –ão [ẽ w̃]. For this reason, allomorphic forms of clitics are also a challenge in monolingual acquisition. They are explicitly taught at school and only fully acquired by literate native speakers of EP. In our study, even some adult monolingual native speakers showed variation in rejecting non-shaped forms after nasal endings. The HSs performed very poorly in this condition and actually showed a strong tendency to accept an ungrammatical non-shaped form. Again, this indicates their reliance on colloquial input, where the difference between *viram-no* and *\*viram-o* is difficult to perceive (see Table 3).

**Table 3.** Morphological shape: Mean accuracy rate, standard deviation (SD), statistical significance (adapted from (Rinke and Flores 2014, p. 690)).

| Condition | Monolingual Speakers (n = 18) Mean (SD) | Heritage Speakers (n = 18) Mean (SD) | Mann-Whitney U | *p* |
|---|---|---|---|---|
| -no/-na (grammatical) | 98.89 (4.71) | 54.61 (24.62) | 20.00 | <0.001 |
| -o/-a instead of -no/-na (ungrammatical) | 82.22 (29.01) | 11.11 (23.98) | 18.50 | <0.001 |

Under the assumption that HSs' linguistic performance can partly be explained by the lack of access to formal instruction, it is expected that their competence may vary depending on the amount of formal instruction they have received in their HL. As mentioned in Section 2, Portuguese HSs frequently attend Portuguese HL classes organized by the network *Ensino de Português no Estrangeiro/EPE* ('Teaching Portuguese Abroad'). Such extracurricular courses take place once or twice a week in the afternoon or on Saturdays for 2 h per week with the main aim to train reading and writing skills in Portuguese. Coming back to the results of the study on the morphosyntactic knowledge of clitics, individual variation can indeed be shown to be a function of the amount of exposure to formal instruction in Portuguese. For the HSs who participated in the study, the years of attendance of the HL course showed a significant positive correlation on their overall accuracy rate (Pearson Correlation Coefficient ($r = 0.652$, $p = 0.006$, cf. (Rinke and Flores 2015)).

Studies on speakers of other HLs in Germany have confirmed these findings. For example, based on evidence from several studies conducted by Kupisch and colleagues, Kupisch and Rothman (2018) reported that French HSs in Germany who have attended a French school in Germany showed a linguistic competence comparable to monolingual French speakers, whereas Italian HSs in Germany who did not have access to formal education in Italian behaved differently from monolingual norms. The study by Bayram et al. (2019) showed a positive correlation between literacy in Turkish acquired as HL by Turkish-descendant second-generation speakers in Germany and more target-like production of passives in Turkish.

## 5. Heritage Languages and Diachronic Change

HS's reliance on colloquial input sources also identifies them as good indicators for ongoing diachronic changes that may not yet be easily detectable based on the standard va-

riety of a language. This has been shown in the aforementioned studies by Rothman (2007) and Pires and Rothman (2009), who demonstrated that inflected infinitives, which are diachronically almost lost in colloquial registers of BP (Pires 2002), are not mastered by BP HSs. Hence, due to their lack of literacy or access to more formal registers, the linguistic competence of HSs may be indicative of previous diachronic changes in colloquial registers.

Another question concerning the relation between HLs and diachronic change is whether HSs can play an active role in initiating or accelerating diachronic change in their HL variety. Theories focusing on diachronic change of syntactic structures often assume that diachronic change happens during the transmission of language, i.e., in first language acquisition. The idea that language change relates to language learning goes back to Paul ([1880] 1920) and has been further developed by Andersen (1973), Lightfoot (1991) and Clark and Roberts (1993), among many others. However, first language acquisition is generally assumed to be a relatively stable and successful process, which leads to the "logical problem of language change" (Niyogi and Berwick 1995): How can language change if child language acquisition generally leads to successful acquisition of the target grammar? To solve this paradox, it is often assumed that certain properties of the input (e.g., previous changes in the frequency of use of certain structures or structural ambiguity) initiate changes in the acquisition of the target grammar (Lightfoot 1991; Roberts 1993). However, the question remains of how ambiguity or changes in the input arise in the first place. One potential candidate is language contact, cf. (Kroch 1999), i.e., a scenario where two languages represent the input for first language acquisition, as it is the case for HSs, who are simultaneous or early successive bilinguals. However, Meisel (2011) called into question whether simultaneous bilingual first language acquisition is indeed a potential source for grammatical change, given that bilingual children "are capable of differentiating the grammatical systems from early on" (Meisel 2011, p. 129). Meisel (2011, p. 121) argued that "simultaneous acquisition of two languages (2L1) typically leads to a kind of grammatical knowledge in each language which is qualitatively not different from that of the respective monolinguals."

We agree with this conclusion, especially for cases in which children receive continuous input and in which the acquisition of core grammatical properties is considered. However, if we include syntactic changes which relate to ongoing processes of grammaticalization, the situation might be slightly different. In such cases, lexical elements (e.g., main verbs) are progressively semantically and phonologically reduced and, as a consequence of this process, can be used as functional elements and in new syntactic contexts (e.g., as modal verbs, cf. (Lightfoot 1979) for the development of modal verbs in the history of English). It is important to note that such developments typically follow predictable diachronic pathways, cf. (Roberts and Roussou 2003). If the specific conditions shaping HL development lead to an acceleration of diachronic changes, the differences between monolingual and HSs should follow these pathways and therefore be systematic in nature.

For example, in our study on clitics in Heritage Portuguese (Rinke and Flores 2014), HSs of EP accepted isolated strong pronouns more easily in the dative case than in the accusative. Because the occurrence of strong pronouns generally requires the presence of an additional clitic, the results indicate that the dative clitic is more easily omitted than the accusative clitic. The more permissive acceptance of ungrammatical datives—which is partly also attested in the monolingual controls—can be attributed to the lower semantic load of dative clitics in comparison to accusatives ones (Andersen 1982). Also, diachronically, datives typically change before the accusative is affected, cf. (Fischer and Rinke 2013; Gabriel and Rinke 2010, with respect to clitic doubling in Spanish). We conclude that the linguistic knowledge of the heritage bilinguals investigated in our study is "innovative, because it promotes linguistic changes which are inherent in the speech of native monolinguals" (Rinke and Flores 2014, p. 681).

Another example from our research on Heritage Portuguese in Germany concerns the distribution of null objects. In Rinke et al. (2018), the spontaneous production of null objects in EP by second-generation Portuguese-German bilingual speakers was compared to first-

generation migrants and two age-matched groups of monolingual speakers in Portugal. One result of this comparison is that the younger bilingual and monolingual groups produced more null objects in their speech than the two older generations, which may be interpreted as reflecting an intergenerational development (Labov 1994) The production of null objects was more expressive in the bilingual group and they also produced significantly more animate null objects than the other speaker groups, as shown in Table 4.

**Table 4.** Rate of animate null objects in the spontaneous speech of four speaker groups (adapted from (Rinke et al. 2018)).

|  | **G1_Bil** | **G2_Bil** | **G1_Mon** | **G2_Mon** |
|---|---|---|---|---|
| animate null objects | 4.1% | 19.5% | 9.4% | 6.6% |

Cyrino et al. (2000) argued that diachronic changes concerning the realization of null and overt pronominal objects follow a referential hierarchy, whereby an extension of null elements proceeds from [−referential] (=propositional, nonspecific, inanimate) to [+referential] (=animate, specific, human) entities. Hence, the higher rates of animate and non-propositional null objects show that the bilingual speakers extend the semantic-pragmatic conditions of null object realization along the referential hierarchy, which corresponds to a language-internal pathway that resembles a diachronic change observed in Brazilian Portuguese, cf. (Rinke et al. 2018).

## 6. Tying Up the Loose Ends

Through the lens of formal linguistics, interested in language systems and the factors that determine them, this paper brings together the results of several studies on EP acquired in Germany as HL and attempts to sketch a coherent picture of the outcome of language development under reduced and less diversified input. The picture that emerges from our research in broad lines echoes the research on other HLs across the globe (Montrul 2016; Polinsky 2018). It represents HSs as native speakers of their home language, whose linguistic competence reflects the influence of external and internal factors that affect native language development in general, i.e., not only heritage languages. Thus, many linguistic particularities, which seem to differentiate HSs' grammars from monolingual homeland speakers' language systems, are the consequence of protracted development of certain properties due to reduced exposure to the target system. Crucially, this protraction does not affect all properties of a language to the same degree. We know that there are many linguistic properties that are instantiated with sparse input and rarely affected in HL development (see, for instance, the use of present indicative morphology in Flores et al.'s (2017a) study on mood selection). It is with linguistic structures that need prolonged, continuous and diversified input that HLs mainly show differences compared to monolingual homeland varieties (e.g., clitic allomorphs, subject pronouns, subjunctive mood). Furthermore, differences are rarely the consequence of sudden innovations but reflect the systematic amplification of linguistic developments already present in the homeland varieties (see (Flores and Rinke 2019) for a discussion of this point).

This leads us to the role of CLI in HL development, which we have not yet addressed. If the language system that characterizes the HL is not fundamentally different from the target homeland language system and if differences can be accounted for by factors that, in principle, may also be visible in monolingual speakers (e.g., reduced literacy, lack of exposure to formal language registers, variation typical of colloquial speech, ongoing diachronic change), what is, then, the effect of the dominant, majority language in HL development? Our claim is that the role of CLI may be much less influential in shaping the heritage grammar than it is often assumed in research on heritage languages (e.g., Benmamoun et al. 2013), especially in the domain of morphosyntax. We show this based on three kinds of observations from previous studies: (i) In some linguistic areas, HSs are not distinguishable from monolingual speakers (i.e., CLI is not a necessary outcome of

heritage bilingualism); (ii) deviations in HLs do not necessarily follow the grammatical patterns of the contact language; and (iii) deviations can be shown to be independent of the contact language if we compare the same HL in contact with typologically different contact languages.

With respect to the first point, we refer to our accent rating study reported in Section 3 (Flores et al. 2017b). This study showed that HSs are perceived as native speakers of EP and, in this respect, contrast clearly with L2 learners. In the study on EP HSs' knowledge of clitics, there were also subparts of the test in which HSs performed like monolinguals. This was the case with respect to topicalization structures, which is per se revealing because topicalization structures are situated at the syntax-discourse interface. Topicalization in EP is an interesting phenomenon because it shows variation: A resumptive clitic may be either used or left out with no change of meaning or pragmatics (cf. ex. 5 from (Rinke and Flores 2014)).

| 5a. | Estes livros, | li**(-os)** | com grande prazer |
|---|---|---|---|
| | these books | read(-CL.Acc) | with great pleasure |
| | | 'These books I read with great pleasure.' | |
| 5b. | Aos colaboradores mais antigos | deu**(-lhes)** | um relógio fantástico. |
| | to the most long-standing workers | gave (-CL.Dat.) | CL.Acc |
| | | 'To the most long-standing workers he gave a fantastic watch.' | |

The results of the grammaticality judgment test (Rinke and Flores 2014) given in Table 5 show that—in contrast to other test conditions—both groups of speakers performed in a very similar way: They accepted topicalization with resumption in general to a higher degree than topicalization without resumption and they showed a higher acceptability of topicalization without resumption in the dative condition. Both results are unexpected from the perspective of CLI as German does not have resumptive clitics and does not show a case asymmetry with respect to topicalization. This case asymmetry is part of the intuitive knowledge of monolingual and bilingual native speakers and was not attested in the L2 learners' results, cf. (Flores et al. 2017b).

**Table 5.** Topicalization: Mean of accuracy rate, SD, statistical significance (Mann–Whitney U-test) (adapted from (Rinke and Flores 2014, p. 692)).

| Condition | Monolingual Speakers (n = 18) Mean (SD) | Heritage Speakers (n = 18) Mean (SD) | Mann-Whitney U | *p* |
|---|---|---|---|---|
| topicalization with resumptive accusative clitic | 97.78 (6.47) | 86.67 (16.80) | 104.00 | <0.05 |
| topicalization with resumptive dative clitic | 91.11 (24.94) | 85.56 (22.55) | 128.50 | 0.179 |
| topicalization without resumptive accusative clitic | 64.44 (28.74) | 58.89 (29.48) | 144.50 | 0.571 |
| topicalization without resumptive dative clitic | 88.89 (17.11) | 80 (24.73) | 129.50 | 0.257 |

Our second claim can also be illustrated with respect to the HSs' knowledge of clitics. As mentioned in Section 5, HSs deviate from monolinguals concerning the acceptance of isolated strong pronouns, permitting it more easily in the dative than in the accusative condition. A general acceptance of strong pronouns could have been attributed to the influence of the contact language, but given the observed case asymmetry, CLI cannot be at stake. Because the same tendency (although to a much lower degree) is also present in the monolinguals, we have attributed the difference between monolinguals and HSs to ongoing diachronic development. A similar conclusion was reached concerning the use of null objects by EP HSs (Rinke et al. 2018). Because German does not license null objects, we would expect a reduction of use of these structures in the HL under CLI—but the contrary

was the case: HSs extended the use of null objects, and they did so in a way predicted by independently attested diachronic pathways.

Our third observation can be illustrated on the basis of two studies comparing different groups of HSs in contact with different majority languages. As shown in Section 3, bilingual children and teenagers show differences in comparison to monolingual children and teenagers, with respect to the interpretation of null subjects in topic continuity contexts, which may be interpreted as protracted acquisition of this property (Rinke and Flores 2018). Interestingly, this delay is attested in German-Portuguese and Spanish-(Catalan)-Portuguese bilingual children in almost identical ways, although the contact languages show typologically different properties (null subject (Spanish, Catalan) vs. non-null subject (German)). A similar conclusion was reached by Flores and colleagues in a study on the acquisition of mood in complement clauses (Flores et al. 2019). Based on a sentence completion task, the study compared mood choice (indicative vs. subjunctive) in Portuguese by two groups of Portuguese-descendant bilingual children: A group living in Germany, with German as the contact language, and another group living in France, with French as the majority language. These contact languages have very distinct mood systems. Whereas in French, the subjunctive mood encodes the same semantic values as EP, the German mood system shows considerable differences. Still, the results indicate that these cross-language differences do not impact on the children's performance. Both groups of bilinguals showed protracted development of the subjunctive, independently of the contact language's mood system.

To sum up, although EP HSs in Germany generally demonstrate very good knowledge of their HL, the acquisition of certain linguistic structures may be delayed due to restricted exposure (e.g., the subjunctive or null/overt subject interpretation). In addition, the primacy of the colloquial register in the input and the lack of formal education may lead to deviations from the standard language (as shown for clitic climbing or clitic allomorphs). The influence of the environmental language seems to be less relevant. In some linguistic domains, bilinguals simply behave like monolingual speakers, although the two contact languages have very different linguistic properties (e.g., concerning accent, or topicalization structures). Some particularities of HLs reflect developments which are clearly different from the majority language (e.g., the dative-accusative asymmetry in EP, which is absent in German), while others can be shown to be independent of the contact language (as shown for subjunctive and null/overt subject interpretation). In sum, the results of several studies have shown that Portuguese spoken in Germany as the HL by Portuguese-descendant second- and third-generation speakers is a native language shaped by its particular input conditions.

In an attempt to relate heritage language development and diachronic change, we suggest that the particularities of heritage grammars may reflect accelerated and extended diachronic changes (for example, with respect to null objects). We think that it is worth addressing the link between diachronic change and heritage bilingualism more closely in the future by combining offline and online measures and relating heritage speakers' linguistic knowledge to their processing of the heritage language in order to investigate the dynamics of heritage language development in more detail.

**Author Contributions:** The authors contributed equally to writing this epistemological paper. All authors have read and agreed to the published version of the manuscript.

**Funding:** This research received no external funding.

**Institutional Review Board Statement:** Ethical review and approval were waived for this article, since this is an epistemological paper which reviews and discusses already published studies. This article does not involve data collection.

**Informed Consent Statement:** Not applicable.

**Data Availability Statement:** Not applicable.

**Conflicts of Interest:** The authors declare no conflict of interest.

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
