# Peer review of "Portuguese as Heritage Language in Germany—A Linguistic Perspective"

_languages, doi:10.3390/languages6010010_

Round 1
Reviewer 1 Report
The paper is a review article, based on the findings from previously published articles from the author(s). Consequently, the questions about the research design and the description of the methods are not fully relevant. While the results have already been published elsewhere, it provides a concise overview of the research conducted so far and draws some interesting and relevant conclusions from its results.
One aspect that could and should be improved is the presentation of the data in the figures. It should be better explained and better contextualized, and the raw frequencies should be given:
Fig. 1: not clear in what amount of data the occurrences of words were detected
Tables 1 and 3: please provide the raw frequencies alongside the percentages
Fig. 2: is the difference between monolingual and heritage speakers statistically significant?
Tables 2 and 4: it is not clear what the "means" given in the table refer to
Other minor remarks:
The abbreviation CLI is explained on page 2, and it is used for the next time on page 10. Please provide the explanation again on page 10.
Page 10, line 352: shouldn't it be positive "correlation", not just "effect"?
At least sections 1 and 2 should be revised (use of tenses, use of prepositions).
Author Response
REVIEWER 1
The paper is a review article, based on the findings from previously published articles from the author(s). Consequently, the questions about the research design and the description of the methods are not fully relevant. While the results have already been published elsewhere, it provides a concise overview of the research conducted so far and draws some interesting and relevant conclusions from its results.
Response: Thank you for this positive evaluation.
One aspect that could and should be improved is the presentation of the data in the figures. It should be better explained and better contextualized, and the raw frequencies should be given:
Fig. 1: not clear in what amount of data the occurrences of words were detected
Response: We added a Table with these data, instead of the Boxplot.
Tables 1 and 3: please provide the raw frequencies alongside the percentages
Response: We took the Tables from the studies that are already published. In the original papers the tables did not include the raw frequencies.
Fig. 2: is the difference between monolingual and heritage speakers statistically significant?
Response: Yes, we added the statistical tests to a footnote.
Tables 2 and 4: it is not clear what the "means" given in the table refer to
Response: We added this information. It means the rate of accuracy.
Other minor remarks:
The abbreviation CLI is explained on page 2, and it is used for the next time on page 10. Please provide the explanation again on page 10.
Response: OK
Page 10, line 352: shouldn't it be positive "correlation", not just "effect"?
Response: Yes, thank you for noticing. We corrected it.
At least sections 1 and 2 should be revised (use of tenses, use of prepositions).
Response: OK (the paper was checked by a native speaker)
Reviewer 2 Report
I very much enjoyed reading this paper. It provides the readers with an extensive overview on Portuguese migration to Germany and the current situation of Portuguese as a HL in Germany as well as its speakers. This is of great importance since heritage Portuguese is only scarcely represented in research on HL in Germany. Especially with regard to the contribution of HL research on the relation between cross-linguistic influence and diachronic change, I find the rationale of the paper quite original.
I have two minor suggestions: The authors might want to devide chapter 2 on "Portuguese migration to Germany" into two subchapters and thus highlight the quite extensive remarks on Portuguese HL education. Also, I would like to suggest that the authors do not use the term "significant effect" when such values are not provided or when the data in question are percentages.
Author Response
REVIEWER 2
Comments and Suggestions for Authors
I very much enjoyed reading this paper. It provides the readers with an extensive overview on Portuguese migration to Germany and the current situation of Portuguese as a HL in Germany as well as its speakers. This is of great importance since heritage Portuguese is only scarcely represented in research on HL in Germany. Especially with regard to the contribution of HL research on the relation between cross-linguistic influence and diachronic change, I find the rationale of the paper quite original.
Response: Thank you for this positive evaluation.
I have two minor suggestions: The authors might want to devide chapter 2 on "Portuguese migration to Germany" into two subchapters and thus highlight the quite extensive remarks on Portuguese HL education.
Response: We followed this suggestion and divided section 2 in two subsections.
Also, I would like to suggest that the authors do not use the term "significant effect" when such values are not provided or when the data in question are percentages.
Response: Thank you, we changed it.
Reviewer 3 Report
Review of Portuguese as heritage language in Germany – a linguistic perspective
In this article, the author(s) present an overview of linguistic research in heritage speakers of Portuguese in Germany. Along with a historical background, the author(s) focus on three factors that have an effect, or not, on the Portuguese of these heritage speakers: (i) quantity of input in the heritage language; (ii) the lack of formal education in the heritage language; and (iii) features of the Portuguese of these heritage speakers as ongoing diachronic development. The author(s) conclude that (i) the quantity of input does have an effect on the delay of some linguistic features (though they are eventually acquired); (ii) the lack of formal education causes these heritage speakers to diverge from their monolingual counterparts; (iii) some features of the Portuguese of these heritage speakers might be an acceleration of diachronic changes present in monolingual Portuguese; and (iv) cross-linguistic influence from German is minimal, if at all present.
A main strength of this paper is that it is well written. The paper was clear and easy to follow. The author(s) do a good job in summarizing the linguistic data to provide evidence for the three generalization that they make at the beginning of the paper. The paper is also innovative, for, as far as I know, the population in the paper has not been extensively studied. I also appreciate the use of “divergent” acquisition rather than more polarizing, and in my opinion wrong, terms such as “incomplete” acquisition.
Minor comments:
- One comment regarding the CLI of German (Section 6). The author(s) state that they do not believe that there is CLI from German, or it is very minimal. The author(s) do an excellent job at arguing for this conclusion. However, I wonder what are the author(s) thoughts on other HL, such as heritage speakers of Spanish in the US. For example, Montrul (2014) looked at the obligatory use of the Direct Object Marker (DOM), a, with animate, specific direct objects in oral production. She found that heritage speakers were omitting the use of the DOM. She argued that, along with insufficient input, transfer from English could have had effect on the omission of the DOM. It has also been suggested that there is a simplification of the grammatical gender system in heritage speakers of Spanish (overuse of the default masculine gender in nouns) due to CLI from English (English does not have gendered nouns). I think that it could benefit the paper to discuss these different conclusions regarding CLI for these two different populations. Could the reason that there is very little CLI from German have something to do with the return-oriented lifestyle? Could it be the effect of the type and quantity of input that they receive when compared to heritage speakers of Spanish in the US?
- I think that the author(s) do a great job in explaining what studies have been done on heritage speakers of Portuguese in Germany. However, I think that the paper could benefit from a brief discussion on the future of this population. That is, what linguistic phenomena should be studied? What are interesting gaps in the literature that must be addressed? Perhaps future steps include the study of the code-switching practices of this population.
- Line 63 – I believe that it should be “In these statistics…”
- Line 98 – I am not familiar with 1970ies when referring to a decade. Usually, just an -s is used, e.g., 1970s. This occurs several times.
- Line 170 – you use the acronym EP without first defining what it is.
Author Response
REVIEWER 3
In this article, the author(s) present an overview of linguistic research in heritage speakers of Portuguese in Germany. Along with a historical background, the author(s) focus on three factors that have an effect, or not, on the Portuguese of these heritage speakers: (i) quantity of input in the heritage language; (ii) the lack of formal education in the heritage language; and (iii) features of the Portuguese of these heritage speakers as ongoing diachronic development. The author(s) conclude that (i) the quantity of input does have an effect on the delay of some linguistic features (though they are eventually acquired); (ii) the lack of formal education causes these heritage speakers to diverge from their monolingual counterparts; (iii) some features of the Portuguese of these heritage speakers might be an acceleration of diachronic changes present in monolingual Portuguese; and (iv) cross-linguistic influence from German is minimal, if at all present.
A main strength of this paper is that it is well written. The paper was clear and easy to follow. The author(s) do a good job in summarizing the linguistic data to provide evidence for the three generalization that they make at the beginning of the paper. The paper is also innovative, for, as far as I know, the population in the paper has not been extensively studied. I also appreciate the use of “divergent” acquisition rather than more polarizing, and in my opinion wrong, terms such as “incomplete” acquisition.
Thank you for your review and the helpful suggestions.
Minor comments:
One comment regarding the CLI of German (Section 6). The author(s) state that they do not believe that there is CLI from German, or it is very minimal. The author(s) do an excellent job at arguing for this conclusion. However, I wonder what are the author(s) thoughts on other HL, such as heritage speakers of Spanish in the US. For example, Montrul (2014) looked at the obligatory use of the Direct Object Marker (DOM), a, with animate, specific direct objects in oral production. She found that heritage speakers were omitting the use of the DOM. She argued that, along with insufficient input, transfer from English could have had effect on the omission of the DOM. It has also been suggested that there is a simplification of the grammatical gender system in heritage speakers of Spanish (overuse of the default masculine gender in nouns) due to CLI from English (English does not have gendered nouns). I think that it could benefit the paper to discuss these different conclusions regarding CLI for these two different populations. Could the reason that there is very little CLI from German have something to do with the return-oriented lifestyle? Could it be the effect of the type and quantity of input that they receive when compared to heritage speakers of Spanish in the US?
We agree that the return-oriented lifestyle and the continuous input Portuguese heritage speakers living in Germany receive in general enables them to successfully acquire their heritage language in many respects. The situation might indeed be different for heritage speakers in the US which show more interindividual variation which may relate to the variable input these speakers receive.
I think that the author(s) do a great job in explaining what studies have been done on heritage speakers of Portuguese in Germany. However, I think that the paper could benefit from a brief discussion on the future of this population. That is, what linguistic phenomena should be studied? What are interesting gaps in the literature that must be addressed? Perhaps future steps include the study of the code-switching practices of this population.
We think that it would be worth addressing the relation between diachronic change and heritage bilingualism in more detail in the future. Moreover, future studies on this population should combine offline and online measures in order to relate heritage speaker’s linguistic knowledge to their processing of the heritage language. We added a few sentences on that at the end of the paper.
Line 63 – I believe that it should be “In these statistics…”
Changed
Line 98 – I am not familiar with 1970ies when referring to a decade. Usually, just an -s is used, e.g., 1970s. This occurs several times.
Changed
Line 170 – you use the acronym EP without first defining what it is.
Changed